# Can There Be Such a Thing as a Sociology of Works of Art and Literary Texts? A Very French Epistemological Debate

**Clara Lévy** [1,*] **and Alain Quemin** [1,2]

1 Institut d'Études Européennes, Université Paris-8, 93526 Paris, France; aquemin@univ-paris8.fr
2 GEMASS—Sorbonne Université/Institut Universitaire de France, 75005 Paris, France
* Correspondence: clevy.paris@gmail.com

**Abstract:** Is it possible to undertake a sociological analysis of works of art? This article considers the arguments for both the negative and positive answers to this question that emerged in France in a vivid manner at the turn of the millennium. It examines the main arguments exchanged by the supporters and detractors of this sub-discipline of the sociology of art, notably those relating to the problem of interpretation (how does one verify this process sociologically when it is applied to a work of art?) and to the ways of presenting evidence. The discussion of these various arguments does not lead to the conclusion that it is impossible to explore heuristically the sociology of artworks in a pertinent way but to an insistence on heightened vigilance and to the formulation of certain principles for the use of sociologists who engage with such objects, if sometimes without the requisite caution.

**Keywords:** works of art; sociology; sociology of art; literature; interpretation





## 1. Introduction

The sociology of art has been characterised by the pioneering role played by French sociologists, who long exercised significant influence over their American counterparts, who in turn helped spread the domain to the rest of the world (Becker 1984; Quemin 2017).[1] It first emerged as a specific domain of French sociology at the turn of the twentieth century with such early contributors as Jean-Marie Guyau (Guyau 1889) and Charles Lalo (Lalo 1927) and, more prominently, in the 1960s, primarily under the impetus of two authors: Pierre Bourdieu and Raymonde Moulin. Bourdieu and his colleagues conducted a seminal body of research into art museums and their publics (Bourdieu and Darbel 1966) that had a lasting influence on subsequent art–public surveys regardless of whether such work sought to affirm or undermine Bourdieu's analyses. Moulin conducted pioneering research into the art market (Moulin 1967) that launched a series of research projects focusing on art markets and institutions. European and, especially, North American scholars fluent in French then disseminated this sociology of art, and often constituted it as a distinct sociological domain in their own countries (Quemin and Boas 2016; Quemin 2017).

So, while 1960s French sociology played a key role in the development of a sociology of art in the United States, and this possibility of undertaking a sociology of works of art had been taken for granted elsewhere, as exemplified by research conducted in the 1980s and 1990s (Griswold 1981, 1992; Peterson 1985; van Rees 1983), the French sociological tradition was long reluctant to explore that possibility. Within the sociology of art in France this debate emerged during a pivotal symposium in Marseilles in 1985 and was revived by Bourdieu in his publications beginning in the 1990s (Bourdieu 1992); it became especially vivid at the end of the 1990s. Since then, the French sociology of art has been confronted with the following questions: is a sociology of works of art possible? And, if so, under what conditions (Fleury 2006)?

Although the debate unfolded beyond the literary domain, this article focuses on that sector of artistic activity because it is the most emblematic of the arguments that initially arose in connection with the visual arts (Quemin 2001).

After first being raised at the 1985 Marseilles conference on the sociology of art, a meeting that would play an important part in the structuring of sociology of art as a proper domain (Quemin 2017), the subject was subsequently neglected.[2] The feasibility of a sociology of works of art was tackled again in Grenoble in 1999 at the first meeting of the CNRS (Centre National de la Recherche Scientifique) OpuS research group (*OpuS* is a French acronym derived from artwork, public and societies—*Œuvres*, *Publics*, *Sociétés*), established to promote investigations in that domain (Majastre and Pessin 2001; Becker 2001).[3] Still, this subject remains a frequent source of controversy, not only between sociologists of art and sociologists of other subfields, but among sociologists of art themselves—the advocates of a sociology of works of art are pitted against those who bitterly contest the very possibility of such a sociological approach (Lévy and Quemin 2007a, 2007b).[4]

Our goal here is to explore whether a literary work can be considered a social statement and a legitimate object for sociological study. Might this field represent a sort of "black box", i.e., a field that defies sociological analysis and winds up as "sociologism"? Should we conclude that since literary works are also studied by other means and by other fields, ones that use "ascientific" approaches, that sociology is ill-equipped to handle this subject? And if not, what contribution does sociological interpretation make? How do we develop a rigorous analytical framework for pursue a sociological approach to works of literature? How do we reconcile the sociologist's interpretation with the authoritative weight of the writer's own—especially when the two seem contradictory?

Moreover, by drawing on examples that take a sociological approach to works of art as well as on our own research, we suggest *if and under what conditions* a genuinely sociological approach (in the broadest sense, i.e., one that offers an explanation and understanding of society) may be applied to texts, one that respects the literary work's other dimensions.

## 2. Results

Although pitfalls await sociologists seeking to analyse literary works, there are certain principles that seem to validate the possibility of performing such an analysis.

### 2.1. The Difficulties of Conducting a Truly Sociological Analysis of Works of Art

In the early days of the sociology of art, Marxism sought to clarify the meaning of the work itself: Georg Lukacs in *The Theory of the Novel* (1920) and Lucien Goldmann in *A Hidden God* (1959) and later in *Towards a Sociology of the Novel* (1964), sought to explain the *content* of works of art within the context of the society that produced them. This theory of structural homology between literary works and then-contemporary social facts is rooted in a sociological perspective that aims to illuminate the socio-economic determinism of all individual behaviour. Goldmann tries to make these structural homologies work at two levels: the general level of socio-economic organization and the cognitive character specific to a particular social group. Still, little of this strand of the sociology of art survives (Quemin 2017). Is this explained by the tenuous nature of trying to develop a sociological approach to works of art? Or, is it due to a general withering of Marxist influence on sociology?

Examining another school of thought together with its chief proponent gives grounds for caution. While *The Love of Art*, published in 1966 by Pierre Bourdieu and Alain Darbel, still appears an admirable work in view of its remarkable advances for the sociology of art, *The Rules of Art*, published in 1992, may be considered a contribution to the sociology of works of art but it also presents interpretive problems (Bourdieu 1992; Bourdieu and Emanuel 1996). Bourdieu's interpretation of Gustave Flaubert's writings was limited primarily to revisiting the plot in its most baldly social dimension to demonstrate the homology between the trajectories of hero and author. Similarly, he systematically made aesthetic assertions based on the homology between social positions and ethical assertions that he rarely situated in their specific stylistic context. Are the limitations of Bourdieu's ideas due merely to the limits of his theory, or more generally to an inherent contradiction in the very idea of a sociological approach to works of art or literature?

Bruno Péquignot notes that a Bourdieusian approach to the sociology of works of art offers one strategy for contextualizing works of literature as repositories of data that may be processed "in the same way as data gathered using other methods" (Péquignot 1999, p. 20). From Péquignot's perspective, works of art are more source or resource than research objects *per se*. Indeed, Bourdieu's sociology of art was, above all, a sociology of cultural producers defined in terms of their positions in the literary realm. As Jean-Louis Fabiani points out, this explains why in *The Rules of Art* "the nub of the argument always revolves around the postulation of a homology between the artwork space and the space occupied by the population of producers. We can always contend that certain types of producers correspond to certain types of artworks" (Fabiani 1993, p. 159)[5]. Consequently, the sociology of art as practiced by Bourdieu when considering Flaubert (a manner of presenting his work that is in itself symptomatic) was more akin to a sociology of production and producers than to a sociology of art works. And yet, Bourdieu clearly hoped to develop a sociological approach to works of art as attested by the bimonthly "sociology of works of art" seminar that he organised at the École Normale Supérieure in the early 1970s (and his more recent, posthumous publication about Édouard Manet) (Bourdieu 2013) over several years.[6] After analysing the trajectory of Flaubert's work in terms of the author's position in the French literary domain, Bourdieu pursued a similar approach for Manet and the visual arts, although this study was less influential—it has received many fewer comments. One of Bourdieu's most brilliant disciples, Gisèle Sapiro, who specialises in the sociology of writers and literature, has recently become more interested in the sociology of works of art. In a recent essay, she examines the content of the works themselves in order to indicate that the practices (for instance, pedo-criminal ones in the cases of the film director Roman Polanski and the writer Gabriel Matzneff) and the values (for instance, antisemitic ones of the writer Louis-Ferdinand Céline) are more easily dissociable from their authors when staged aesthetically; none of Polanski's films deals with paedophilic relationships, unlike Gabriel Matzneff, who obligingly reports on his own paedophilic practices and relationships (Sapiro 2020). Nonetheless, Sapiro seldom devotes her attention to the sociological analysis of visual art.

In spite of a renewed interest in this subject in the late 1990s, the French author Nathalie Heinich wrote a scathing assessment of the sociology of of art: "The sociology of works of art constitutes the most obvious, controversial, and probably the most disappointing aspect of the sociology of art" (Heinich 2001, p. 87). What can explain her so severe judgment although the very existence of the brilliant work of Wendy Griswold in the United States should have made it impossible to formulate such criticisms (Griswold 1981)?

First of all, the problem of interpretation appears of key importance in the sociology of art. In sociology, there is generally a minimum consensus around any proposed interpretation. For example, when confronted with a phenomenon such as social mobility, sociologists specialised in the field will generally agree on the need to use a social mobility table. And in the analysis of said table, there is a second, widely accepted consensus that the diagonal reflecting the phenomenon of social immobility is especially relevant. Obviously, these could still produce a range of interpretive differences that could generate diverging analyses linked to particular schools of thought. However, such divergences are largely attributable to theoretical choices that are clearly held and defended by their proponents. Not so in the sociology of works of art. Here, interpretation appears much more unstable with a risk of infinite personalisation. While this is partly a difference of degree, it nevertheless introduces a specific characteristic that tends to set the sociology of works of art apart from other legitimate objects of sociological study[7]. In order to validate interpretations—or even to offer interpretations in the first place (some of which may differ, or even contradict the interpretations of the artistic creator and his/her public)—the sociology of works of art must agree that art is a language whereby "the existence of a code would enable the sociologist to dispense with having to refer to the interpretations of the artist and his/her public" (Raynaud 1999, p. 124).

Thus, legitimacy can be granted to the sociology of literary works. As a matter of fact, literary and narrative art forms are better suited to sociological analysis than forms that permit—or invite—a far greater projective dimension to the analyst's interpretations[8]. It is probably no coincidence that Dominique Raynaud's strident criticism of the sociology of works of art uses the example of the visual arts (Raynaud 1999, p. 119), and Nathalie Heinich, while remaining lukewarm *vis-à-vis* the sociology of works of art in general, concedes that the greatest success in this domain have been made in the sociology of literature.

But, while it is only *a* sociological interpretation, is the fact that it is not *the* sociological interpretation not clearer than in other domains[9]? If we take a particular literary text (or film, to choose another narrative form of art), would another sociologist wishing to analyse the same work from a sociological standpoint have identified the same significant elements? Would they produce the same analysis (or, at least, an analysis sufficiently similar so as not to be attributable merely to the psyche of the analyst)? Advocates of the sociology of works have confronted this question[10]. Some suggest solutions that lead to a sociology of reception rather than to a sociology of works of art *per se*[11]. A productive experiment might entail several sociologists independently analysing the same work of art—especially a literary text—followed by a collective examination of the various analyses. After all, it is to be hoped that such an exercise would help to advance the debate regarding whether works of art can be analysed from a sociological perspective. Obviously, one could not expect complete agreement among the various analyses of the works of art—and in any case this is not demanded in other well-established fields of sociology. Nonetheless, strong divergences in the analyses would question the scientific nature of the approach, particularly its interpretive dimension.

Instead of opting for an internalist approach that seems insufficiently grounded from a sociological perspective—given that the works are decontextualised from the social framework from which they emerge—we offer a broadly external perspective. If any work of art and especially a text is to be situated in a social perspective in order to give sociology something to say, the sociologist must allow the creator sufficient freedom for the work of art to exist as an independent aesthetic expression. The notion of "information" provided by the work is undoubtedly a more effective way of doing this than other more mechanical approaches and that information has to be interpreted and analysed by the sociologist.

Finally, is it possible to deal with a single literary work from a sociological perspective? When Jean-Claude Passeron uses convincing epistemological arguments to demonstrate the impossibility of analysing a single individual case from a sociological standpoint (Passeron 1991), why should it be any different for a single work of art? Similarly, does a sociological examination of a series of works by *a single* writer or even his/her entire *oeuvre* offer the possibility of avoiding the aforementioned pitfalls?

If one accepts the possibility of examining a single literary work from a sociological perspective, this would constitute only one possible approach. It would shed light on only a small part of the work and neglect its full significance. This harks back to the theory of Jean-Claude Passeron concerning the pleasures of art (Passeron 1990, pp. 99–123), who asserts that what defines a work of art is its multiplicity of signifiers. Surely this is proper domain of the sociology of works of art. Consequently, analysing a work of art implies harnessing a wide range of sociological as well as non-sociological approaches in order to enhance the audience's appreciation of *the work itself*.

After all, a sociology of works of art—in particular literary works, but also films, or any art form that bears a narrative dimension is possible only if undertaken in a very careful manner. (There is a significant French tradition of the sociology of music, but no noticeable sociology of musical works strictly speaking, probably precisely because of the absence of explicit narration.) This then becomes less a sociology of a work of art—in the sense of a sociology of education or a sociology of the family—than a sociological *perspective* on the works that does not claim to unlock all of their significations or even to constitute a central viewpoint. A sociology of literary works is possible only when it focuses on the

production of such works and continues with an analysis of the conditions under which they are consumed. And yet it is paradoxical that the sociology of works of art reveals little about the work itself *per se*, i.e., about its intrinsically artistic dimension. When this artistic, or literary, dimension is questioned epistemological problems arise. These must be identified and dissected in order to provide a genuinely sociological analysis of works of art.

*2.2. The Real Possibility of Developing a Sociology of Works of Art*

By the 1985 Marseilles conference, the French sociologist and epistemologist Jean-Claude Passeron had already traced the path he considered indispensable: "we have long been patiently waiting for the sociology of art to fulfil the dual pledge inherent in its title, i.e., that it takes its rightful place as a sociological sub-field by making an intelligible contribution in a similar format to other sub-fields, and *also* that such sociological knowledge consists specifically of knowledge of the works analysed as works of art *per se* and their effects in terms of aesthetic affects" (Passeron 1986, p. 449).

For logical reasons, sociology can focus on works of art—especially literary texts—*stricto sensu* while remaining within its academic boundaries. Clearly, a literary work is a social object as well as an aesthetic statement and can therefore be analysed by sociology. The tricky question of the interpretive freedom of sociologists of works of art—that only arises in this sub-field—is prompted by a scientific conception of sociology that ignores the vulnerability of many sociological interpretations in all domains.[12]

Not even the fiercest opponent to the sociology of works of art would deny that a literary work is a social statement[13]. This implies that art works represent a legitimate topic for sociological study, just like any other social domain. This implies a shift in the initial question, which no longer concerns the sociologist's legitimacy to conduct his/her research into the artwork but to focus on a particular sociological approach to treating works of art as social elements. The question then becomes one of method: how to produce specifically sociological knowledge relating to the work of art, i.e., knowledge that is neither purely formalist nor excessively sociological.

The key importance of the distinction between the form and function of the literary object was first highlighted by Russian formalists[14], particularly Y. Tynianov who stressed both this difference and the inseparable nature of substance from form within the literary text. This is one of the main characteristics used (especially in schools—hence the insistence upon the additional substance/form dichotomy in literary works) to analyse the specific nature of literary texts: writing practices are not subject to the expression of a pre-established thematic but entwined within the literary text. Once the relationship between substance and form has been recognised within the text, sociology's contribution *vis-à-vis* formalism would consist in investing this relationship with a sociologically relevant meaning, by inserting it within the logic of the social world in which the work was written. As Jean-Claude Passeron explains:

> It is generally agreed that the sociology of works of art can only exist if it is able to explore the relationship between the structures of the works of art and the internal functions of these elements, and the structures of the social world in which their creation, diffusion and reception actually meant something or served some function.

This entails opposing externalist sociologism, insofar as the analysis of the social effects or contexts of art involves an internal structural analysis of the work of art. An analysis of an isolated work runs the risk of nullifying itself if it attempts to serve as a catch-all approach to analysing a symbolic practice rendered anonymous. In addition, it means opposing rampant internalist formalism insofar as the internal analysis of 'literarity' or pictorial iconicity must find sufficient reasons and relevant issues within the structure of the text to guide the external analysis of the functioning of the work that is akin to a cultural functioning" (Passeron 1986, pp. 455–56)[15].

The sociologist may benefit from this dual advice. Take, for example, a work that is considered emblematic in this regard: *A Void* (*La disparition* 1969) by the French writer Georges Perec. Form and function vie for attention in an exaggerated manner that can be analysed fully only once it is related to the author's autobiography. *A Void* is a lipogrammatic novel, i.e., one that uses a constrained writing technique consisting of omitting a letter or group of letters, in this case "e", the most common letter in French. So, while we could read the entire novel as a "normal" work without being aware of the complete absence of the letter "e" throughout its 326 pages in the French version of the text, once this constraint is revealed, everything takes on a new aspect—not only in terms of the formal constraint that has now been identified, but also in terms of the novel's plot which is developed around this absence. One of the main characters, Anton Vowl[16] (atonic vowel), suspects that his speech has something missing, and Perec identifies what this is in the third page of the book—«*un rond pas tout à fait clos, finissant par un trait horizontal*» ("a circle that is not fully closed, that finishes off in a horizontal line"). The novel comprises twenty-six chapters, but the fifth one is missing (just like the fifth letter of the alphabet); the sections are also numbered—one to six—but the second is missing (like the second vowel). What is of special interest is how the link between substance and form can function ideally in the composition of a literary text in which the formal constraint sets the tone as well as the background for the plot, as the book itself constantly explains how it came to be written:

> "Word after word emerged, black on white, fired from a cannon as arduous as it first appeared insignificant for those who read without knowing the solution to a novel that, strange and all as it seemed, suddenly appeared to him to be rather satisfactory ( . . . ).

And then later on, when he was surer of himself, he lent his narration a symbolic touch which first followed the line of the novel to the letter, reconstituting and revealing it without ever betraying completely the Law that had inspired it. This Law enabled him—not without friction and occasional bad taste, but neither was it devoid of a certain humour and brio—to tap into a rich productive vein that stimulated his innovation to the highest degree." (Perec 2008, pp. 310–11 (our translation; in the French version of the text, the letter *e* is systematically avoided)).[17]

However, the importance of the theme of identity and the question of Perec's Jewishness could go unnoticed (as it did for a long time) if the sociologist refused to link the stylistic device of omitting the letter "e" with the disappearance of "*eux*"[18]—i.e., the death of his father at the front and the deportation and death of his mother as a non-French Jew, together with all of the Jews exterminated during the Second World War—which Perec aestheticised in this work. His work constitutes a social statement that can only be understood in a specific socio-historical context. The overriding importance of this context in analysing the work of art cannot be appreciated unless the sociologist takes account of it in a very specific manner in the work's form. The next stage in understanding the social statement constituted by the book is to reinsert the work into the context of the author's oeuvre, and then within the body of works produced by other contemporary writers, in particular Francophone Jewish ones (since they share social characteristics with Perec) in order to determine, in a comparative sense, whether the aestheticisation of the theme of identity encountered here operates on a broader level (Lévy 1998). Of course, other types of literary analysis also consider the biography of the author and relate it to elements of their work. But the contextualization of the work of *several* authors in relation to each other, at a given time, in a particular literary space, is very rarely carried out by specialists of literary analysis except precisely when they mobilize a sociological perspective. The role of the sociologist of art (more specifically here, literature) consists in defending the adoption of this perspective in literary studies.

The issue of sociology's interpretive margin evokes one of the fiercest criticisms levelled at the sociology of works of art: how can sociology provide a scientific analysis when such an analysis is based on an interpretation that is frequently difficult to control? How can we be sure that the proposed interpretation would be acceptable to the artist who

created the work? What criteria should we use in deciding whether this interpretation is a valid one (or—even more ambitiously—the correct one) in relation to the myriad other interpretive possibilities?[19] These reservations underpin Florent Champy's study based on multiple sociological interpretive analyses of the works of another French writer, Marcel Proust. We cannot fail to note "the enormous diversity in explanations of how Proust perceived society which are variously presented as interactionist (Belloï), as a precursor of Bourdieu (Dubois), as a theorist of social change in a similar vein to Bourdieu and Elias (Bidou-Zachariasen), as a disciple of Tarde who was sceptical of history in older Proustian studies (Henry), or Durkheimian for the philosopher Vincent Descombes. How can a work of art—even a literary work—simultaneously contain sociological messages that are difficult to reconcile from a research perspective when they are based on such divergent perceptions of the individual? What is the point of the sociological interpretations proposed, and what project-specific criteria can be used to evaluate their respective relevance?" (Champy 2000, p. 347).

This criticism of the excessive importance accorded to the interpretive aspect of analysis should apply to sociology as a whole and not just to the sociology of works of art. The issue of interpretation arises frequently in sociology, even in sub-fields that lie at the heart of the discipline, and even for arguments built on strictly quantitative data. An iconic example of this is the bitter controversy over the interpretation of unequal opportunities in France in the 1960s and 70s (Bourdieu and Passeron 1964, 1970; Bourdieu et al. 1977, 1979; Baudelot and Establet 1971; Boudon 1973). Several seminal works began with identical data and the same observation: schools did not fulfil their designated role of promoting upward social mobility, and even helped to perpetuate social inequalities from one generation to the next. Proceeding from this observation, authors offered radically divergent analyses founded on conflicting epistemological premises and theoretical options, and differing interpretations of social reality. A Bourdieusian interpretation stressed the role of structures, in this case schools and teachers. A Marxist interpretation highlighted the importance of the bourgeois ideology that the two school systems instilled in children from bourgeois families, on the one hand, and working-class children on the other. Finally, Boudon's interpretation underlined individual agency and the choices individuals make in choose taking stands and comparing the costs and benefits of a given situation. This example merely shows the extent to which the interpretive analysis of statistical data, guided by an author's theoretical stance, can give rise to radically different analyses of the same research object.

The contested question of interpretation becomes more acute when sociology delves into the qualitative sphere, for example, when studying the gap between theory and practice. The American sociologists D. Roy and M. Burawoy, disputed the theories of C. Taylor, and demonstrated that workers work more than required from a rational perspective, and more than they are willing to recognise, because they allow themselves to be seduced by the mass production "game" (Fournier 1996, pp. 88–93). Based on participant observation of assembly-line workers at a thirty-year interval, researchers noted that workers did not attempt to slow their pace of work. Significantly, these sociologists noted that their findings could not have been gleaned on the basis of interviews alone. Here again, an interpretive is required to explain why and how under-paid individuals work more than they must and why they rarely admit this fact to outsiders. The interpretation advanced by Roy and Burawoy (turning their work into a game) does not completely invalidate Taylor's (systematic time-wasting).

The question of interpretation becomes even more salient in the sociology of works of art because it relates directly to the production of proof discussed earlier—paradoxically this is one of the only sub-fields that provides readers with an analysis that can be compared directly to what is actually being analysed, i.e., the work of art in general or even more literary. This represents an exception in the sociological sphere, where the sociologist's data are usually presented in filtered form (and thus partially deformed—and always deformed in a way that make them compatible with the argument proposed). Often, even

when sociologists include a comprehensive appendix, they do not consider every element composing the social reality that they are studying. Indeed, before analysis, data are filtered to make it compatible with the research, by producing statistics, conducting interviews, or by field observation. In the case of the sociology of works of art, it is particularly easy to access such material.

Without the artist's—or the writer's—input to validate the sociological interpretation, it is easy to challenge sociological findings on the grounds that it is impossible to ascertain the reliability of one interpretation over another, and that all interpretations are potentially fanciful and prone to the subjectivity of the researcher. This challenge to sociological interpretation appears even more compelling when it comes directly from the author. And yet in what other social sphere do we submit sociological arguments for validation by the social actors being studied? The answer is, virtually none, apart from researchers directly involved in the sociology of action where the resistance of certain individuals faced with a sociological analysis of their position may be stressed—'proof' of the validity and relevance of the analysis that triggered the emotional response. By the same token, one can counter Heinich's argument—"it is not the work of art that teaches us most about the artist's intentions, but the artist him/herself insofar as they can explain their own work" (Heinich 2002b, p. 135)—by considering that it is not necessarily the artist's intentions as explained by him/herself that teach us more about the work, but the work itself.

In more general terms, in response to the criticism of the interpretive component of sociological analysis, Passeron's rebuttal –applicable to all branches of the social sciences— contends that when it is impossible to provide incontrovertible proof, we have to "make do" with a bundle of converging indicators (Passeron 1991)[20]. In other words, sociology must come to terms with its undeniable interpretive dimension by offering sufficiently robust and straightforward arguments (e.g., by providing multiple explanations of elements that have led the sociologist to choose one interpretation over another) in order to enable the reader to reconstitute the author's line of reasoning and to decide, in full knowledge of the facts, whether to adhere to the conclusions. As Heinich points out, based on her interpretive analysis of a novel by Ismaïl Kadaré, "a text can lend itself to multiple interpretations— and it is perhaps this very capacity that is (partly) a hallmark of its quality—without one necessarily invalidating another" (Heinich 2002a, p. 217).

If, instead of striving for a univocal approach, sociological reasoning opted for a plurality of interpretations with a certain degree of convergence, one can envisage a sociology of works of art under certain conditions. To revisit the example of the sociological interpretations of Proust, Florent Champy demonstrates that all interpretations are plausible if somewhat irreconcilable (just like the theories of unequal opportunities discussed previously), because they are founded on analyses of certain passages of Proustian texts provided by the bundle of indicators to which Passeron refers. The hazardous part of an interpretative approach can be addressed by considering Luhmann's ideas concerning the communication system operating in works of art. Indeed, "the work of art . . . is only born thanks to a network of recursive relations with the other works of art, with frequent verbal communications on the art, with technically reproducible copies, exhibitions, museums, theatres, buildings, and so on" (Luhmann 1995). The work of art only exists—and can therefore only be understood—in connection with other works, within the social, historical and material context in which it appears and by considering the discourses surrounding it.

## 3. Conclusions

Although the debate concerning sociological approaches to interpreting works of art lost some of its vigour in the past decade, it is important to revisit issues raised by this approach from an overall perspective and—in spite of the epistemological difficulties more or less specific to this type of research object—to illustrate ways in which it is possible to conduct sociological analyses that shed new light on what may be described as "the work itself". We can also consider works of art, in particular literary works, as aids to understanding the social world and therefore a welcome stimulation to the sociological

imagination (Barrère and Martuccelli 2009). If sociology has made a decisive contribution to art appreciation by pointing up the social dimension—from production to reception, it is still possible—by proceeding carefully and with greater ease in certain domains of the arts than in others—to adopting a sociological approach to art that enhances our understanding of it. But this will only be possible if we take account of the specific characteristics of the object in question, as well as to the best way of moving forward more confidently using an approach based around the sociology of works of art. This means trying to avoid an over-zealous belief in absolute sociological objectivity while not giving up on getting as near to it as possible. The sociologist must strive to ensure that their approach to the work facilitates its understanding, but even more importantly that they contribute an explanation of society or social facts. This also involves claiming and coming to terms with what some sociologists of art refer to as "the exorbitant entitlement to choose a point of view" (Hennion 2002, p. 221). But is this exorbitant entitlement not given to every sociologist regardless of his or her chosen subject, so long as the point of view is "reasonable" and scientifically defensible? The sociologist of art must clarify the standpoint from which they produce their findings in an honest way and recognise that the goal is not to universalise these findings but to submit them for scientific, collective discussion. Thus, recognising that the sociology of works of art is possible under certain conditions would undoubtedly help to bolster the status of the sociologists committed to this domain. We believe it is important to involve sociologists from other countries (Quemin 2017) in a debate that has become recurring withing the French sociology of art. Clearly, the best way of moving forward in this area is through theoretical reflection on the topic itself and the practice of the sociology of works of art in various cultural contexts and other national sociological traditions.

**Author Contributions:** C.L. and A.Q. contributed equally to this manuscript. All authors have read and agreed to the published version of the manuscript.

**Funding:** This research received no external funding.

**Conflicts of Interest:** The authors declare no conflict of interest.

## Notes

[1]   It should be stressed that before that, sociology of art was much developed by Eastern European authors as early as the 1920′s for Hungarian sociologist Georg Lukacs (Lukacs 1963, 1974) or the 1950′s for his follower Romanian born and raised Lucien Goldmann (Goldmann 1964, 1970, 1975, 1976a, 1976b, 1976c). Still, their influence on the present sociology of art remains limited today.

[2]   When commenting upon the Marseilles conference in the postface of the proceedings of subsequent workshops organised in Grenoble in 1999, Raymonde Moulin, one of the major figures of art sociology in France, concluded with a question: "is a sociology of works of art really possible?" (Moulin 2001, p. 465).

[3]   In the chapter devoted to the sociology of art of a 1988 publication (Mendras and Verret 1988), Raymonde Moulin did not deem it necessary to deal with the sociology of works of art, preferring to focus on other themes (Moulin 1988, pp. 185–93). In a 1999 reissue of a dictionary of sociology (Boudon et al. 1999), she devoted a single—relatively critical—sentence to the matter (Moulin 1999). The introduction of the issue of the journal "*Recherches sociologiques*" (sociological research) devoted to the sociology of art—taking up where the Marseilles conference had left off—wondered whether "a sociology of works of art is really possible?". The papers included in the issue did not really provide an answer -neither in general terms nor by analysing case studies (Dechaux and Ducret 1988, pp. 129–32). In the chapter on "the sociology of art and culture" included in *La sociologie française contemporaine* (Berthelot 1999, pp. 251–63), another French sociologist of art, Bruno Péquignot, devotes the last of the six points that he raises to what he terms "the science of works of art" and points up numerous recent developments of the domain. In France, along with Olivier Majastre and Alain Pessin (Majastre and Pessin 2001), Bruno Péquignot has been one of the most fervent promoters of the sociology of works of art (Péquignot 1997, 2001, 2003).

[4]   Among more recent publications, we should mention Péquignot (2007) and Jean-Pierre Esquenazi (2007) which are notable for the importance accorded by the authors to the question of artworks: the former highlights "the question of artwork in the sociology of art and culture" (although Bruno Péquignot recognises that this deliberately-chosen title only covers part of his article 2007, p. 25) while the latter, "Sociology of artwork", corresponds both to the general title of the publication and to a chapter.

[5]   Nevertheless, we should note that certain authors have succeeded in developing highly convincing sociological analyses of artwork even while maintaining a relatively strict Bourdieusian perspective (Héran 1986, pp. 317–34).

[6]   We would like to thank Jean-Louis Fabiani for providing us with this information.

[7]    Nathalie Heinich pursues this line of argument by insisting upon "the absence of any method for providing a sociological description of works of art—apart from the descriptions of the actors, which leads us back to a sociology of the reception of works of art" (Heinich 2001, p. 88).

[8]    And within given fields of artistic expression, the sociological analysis of abstract art would probably be riskier than figurative art, poetry riskier than novels, etc.

[9]    As we are aware, since Max Weber, the question of interpretation is an especially hot topic in the social sciences (see *inter alia* Weber 1949, 1992) as experts' subjectivity and their stance in relation to different values are inevitably expressed with great vigour in their analyses (Boudon 1999, 2001). Nevertheless, in a number of sub-fields of sociology such subjectivity and values may be more easily controlled than in the sociology of art, and particularly the sociology of works of art.

[10]    "How can we avoid completely "ludicrous" interpretations of a work of art, with absolutely no "safeguards", like we often encounter in articles reviewing or presenting exhibitions—ridiculous caricatures that are at best useless, and at worst, harmful?" (Péquignot 2007, p. 207).

[11]    This is the case for Jean-Pierre Esquenazi, for whom "the issue of the meaning of a work of art bypasses its producers and passes on to its interpreters. It is not the work's creators who decide or forge the meaning, but the users or—put another way, the different publics" (Esquenazi 2007, p. 83).

[12]    Obviously, provided that in this sub-field, as in other sociological sub-fields, convincing and much less convincing research is carried out.

[13]    Bruno Péquignot reminds us of this point: the sociology of art is primarily a branch of sociology, i.e., it seeks to produce knowledge of social matters (2007, p. 17).

[14]    "*Formalism*, the term originally used by the adversaries of the movement, was the word used to describe the school of literary criticism prevalent in Russia between 1915 and 1930" (Todorov 1965, p. 15, and Todorov 1988).

[15]    We should also note that this comment by Passeron also provides an answer to the question—based on an observation by the same author—concerning the possibility of analysing a single work from a sociological perspective.

[16]    The French term for vowel is voyelle and contains two e's.

[17]    Alors naquit mot à mot, noir sur blanc, surgissant d'un canon d'autant plus ardu qu'il apparaît d'abord insignifiant pour qui lit sans savoir la solution, un roman qui, pour biscornu qu'il fut, illico lui parut plutôt satisfaisant ( . . . ).Puis, plus tard, s'assurant dans son propos, il donna à sa narration un tour symbolisant qui, suivant d'abord pas à pas la filiation du roman, puis la constituant, divulguait, sans jamais la trahir tout à fait, la Loi qui l'inspirait, Loi dont il tirait, parfois non sans friction, parfois non sans mauvais goût, mais parfois aussi non sans humour, non sans brio, un filon fort productif, stimulant au plus haut point l'innovation.» (Perec 1969, pp. 310–11)

[18]    In French, the phrase "*sans e*" ("without e") sounds nearly similar to "*sans eux*" ("without them").

[19]    Dominique Raynaud considers this to be one of the many obstacles to a scientifically valid sociological approach to works of art (Raynaud 1999, p. 129 *et seq*). In his view "the uniformity of interpretations—or at the very least a low degree of variability—is a necessary condition for ensuring the logical consistency of a sociological approach to works of art".

[20]    Bruno Péquignot has attributed the image of the interpretation of indicators to Carlo Ginzburg who developed it in the 1980s (2007, p. 273).

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
