# Peer review of "Can There Be Such a Thing as a Sociology of Works of Art and Literary Texts? A Very French Epistemological Debate"

_arts, 2022_

Round 1

Reviewer 1 Report

First, I would like to point to an unclarity. It seems that in the text the terms 'sociology of art' and 'art sociology' are treated as having a particular meaning, the former implying the sociology of works of art, the second the general sociological sub discipline. But at many points in the text I am simply not sure which is meant. For me, using 'sociology of works art' consistently would be useful, as in the literature I am used to, the terms 'art sociology' and 'sociology of art' are simply synonyms.

Second, the introduction of the text seems to point towards providing an overview of the French debate about the possibilities of a sociology of art works and objections to it. However, sections 2 and 3 of the text, present a description of that debate that seems to be thematically structured. From the text, I cannot see how that structure has been developed. More importantly, this leads to a structure where the authors voice their opinion on arguments presented by others throughout the text. For me, as not well-versed in the French debate, it would have been more interesting to first read a chronological description of the French debate with a neat delineation of the arguments pro and con a sociology of works of art. Only then, it would be logical to read the authors opinions on the matter (that I do not object to by the way). 

Third, at many points in the text I am left wondering how a sociology of the works of art as the authors seems to view it would be, would differ from art historical analyses or literary scientific analyses. This occurs when examples of works such as the Perec novel are discussed. I fail to see what the additional value of a sociological analysis of the work would be over a literary analysis that pays due attention to the author's biography, as I think it must and frequently also does. Or are the authors referring to a particular type of literary theory here? A type that in my view, would simply be very limited.

Fourth, for a reader familiar with the art sociology of Niklas Luhmann some of the arguments in the paper are simply odd. Particularly the strict distinction that is made between formal analysis of works of art and the content or meaning of the works, is surprising. The authors do not agree with the distinction, luckily I would say. Nonetheless, this makes a lot of the arguments raised irrelevant to me. This points to the fact that no consistent attempt to relate the debates in French sociology to the debates outside of the country is made, although some references to American debates do occur. Why then, not also refer to this more German-based tradition that has, by the way, been further developed by e.g. Flemish and Dutch writers in art sociology such as Laermans, Gielen and Van Maanen to point at just a view?

Therefore, I am inclided to think the goal of the article is to present the French debate to an international audience. A goal that I would wholeheartedly support. However, the manuscript fails to do this consistently. The text seems undecided between presenting the French debate on the matter and weighnig in on that debate. If the sketch of the debate is concise, necessary relations to debates outside France implemented consistently, and then the authors present their own view on the matter, this article could work. But not in its current form.

Author Response

Reviewer 1

We would like to thank this reader for the quality of his /her proofreading and for the relevance of his / her  pieces of advice, which we have almost all strictly followed.

1- We have followed the first recommendation, to remove any ambiguity between "sociology of art" and "sociology of artworks".

2- We have explained the difference between the sociological analyses of literary works and strictly literary analyses: as a matter of fact, those, among the latter, when they take into account the biography of the author and the socio-historical context, mobilize precisely the sociological perspective.

3- We have introduced a reference to the analysis of N. Luhmann.

4- On the opposite, we chose not to totally overhaul the general structure / construction of the article, in terms of its development and the succession of arguments, as the other two referees did not share that criticism and both validated the construction of the article with not any limitation.

Reviewer 2 Report

Interesting paper which discusses in a sometimes ludicrous way the potentials and aporias of French art sociology in developing a convincing approach of works of art and which supports the idea - as stated in the short conclusion - that this debate should be reopened and this possibility explored again in a thorough manner. Very well written and plain in its structure, the article has only few possibilities of improvement in our eyes:

-at the beginning of the text, the distinction between "sociology of art" in a general sense and notably related to life style, audiences, markets (where France, via Bourdieu, and Moulin in the 1960s but also Edmond Goblot in 1925 who is surprisingly not cited here, played an important role) and “sociology of works of art”, which is the actual object of the paper, is not clear enough. We propose therefore to say, at line 34: "...the very possibility of undertaking a sociology of works of art had..."

-also, Laurent Fleury's Sociologie de la culture et des pratiques culturelles (2006), who discussed this very French connection between art sociologie and policies, could be cited at the beginning of the text

-the timing is not always very clear: why is the debate described as being ever-present "since the late 1990s" (line 39) while the question of a sociology of works of art was already prominently arisen in the 1986 Marseille colloquium book (as the article also later states) and Bourdieu's contribution dates back from 1992?

-in the subtitles 2.1 and 2.2 the reference to a "truly" sociological analyses and to a "real" possibility of developing it can raise questions (what is a 'true' or 'real' sociology?), but it may also be understood in a playful manner as mentioned and would then be ok

-speaking of the structure, there is a section 3 missing (no element between 2.2 and 4. Conclusion, which should probably be numbered 3)

-rare formal or layout problems: at lines 254 (in the brackets) and 318 (before "How can a work...") there seems to be a space too much. Also, the Perec citation (lines 281-289 and footnote 19) seems to have an odd layout (unequal margins of both text portions)

 -finally, it is not clear if there have or not been interesting contributions to this debate since the turn of the millenium, when almost the totality of the relevant references of the article stop. This is all the more so as the conclusion states that the topic "is still debated in France today". It then would be beneficial for the reader to discuss at least some recent references

-one of them - to limit oneself to the literary field focussed by the text - could be "Le roman comme laboratoire" (Anne Barrère and Danilo Martucelli, 2009), which was only recently (January 31, 2022) presented by Anne Barrère and discussed at a séminaire of the CERLIS/Sorbonne nouvelle organized by Olivier Thévenin and Bruno Péquignot

-a more personal input: I always find it not ideal when a text ends with a reference (here of 2017), which could induce the idea that this analyses only repeats what has already been said some years ago. As the reference has already been referred to at the very beginning of the article, this last reference could also be dropped or at least put 1-2 sentences up to be less prominent.

Author Response

We would like to thank this reader for the quality of his /her proofreading and for the relevance of his / her pieces of advice, which we have absolutely all strictly followed.

1- We have introduced a reference to the work of L. Fleury.

2- We have clarified the timing of the debates on the sociology of works of art in France.

3- Indeed, as the proofreader assumes, the terms "real" were impregnated with irony.

4- We have re-numbered the parts and corrected our mistake.

5- We have corrected the few formal presentation errors that were pointed out.

6- We have introduced references to more recent research (for example, as suggested, the work of A. Barrère and D. Martuccelli).

7- We have removed the final reference.

Reviewer 3 Report

Very well written, with strong arguments. It is an ambitious paper, but it isn't easy ro respond in a short text  to some so large and hard questions:  "Is a literary work a social statement and is it a legitimate object for sociological study?"How can sociology provide a supposedly scientific analysis when such an analysis is more or less wholly based on an interpretation  that is frequently difficult to control?"; "Is a literary work a social statement and is it a legitimate object for sociological study? Does this whole area not represent a sort of “black box”, i.e., a field that defies sociological analysis and almost inevitably winds up as “sociologism”?", etc. The author' main objective is to "attempt to demontrate if and under what conditions a genuinely sociological approach (in the broadest sense) is possible". That a really good question, and l like his argumentation which is "nuancée".

-The author speaks aloso about vsual arts ans in his bibliography that are more refreneces to the sociology of visual arts than to the literature Pequinot, Quemon, etc.). And why nothing about music. There is in France a tradition of good sociological analysis of music: Pierre-Michel Menger, Antoine Hennion. l understand it is impossible ot speak in a short paper to speak about every art.

Other few comments:

-The author presents, what it is true, Lucien Goldmann as the pionneer of sociology of art in France with. 5 references to his works in the bibliography but no presentation of his perspective in the paper.

-Pierre Bourdieu is a central figure of the contemporary French sociology. In the bibliography there are many references to his works. In his paper, the author presents shortly his first works (with Darbel ,etc.), his Rules of the Arts, with a short and severe critic  and   his posthumous book on "his" Manet, but without any  comment.  Why not few words about this important book? And l don't think we need in this papaerthe paragrah about the  controversy over the interpretation of unequal opportunities in France in the 1960s and 70s between Bourdieu and Boudon.

-In the bibliography, there is no reference to the Gisèle Sapiro's work, a prolific author, and to her most recent book, Peut-on dissocier l'oeuvre de l'auteur? (2020).

My conclusion is : the paper should be publish but with many minor corrections.

Author Response

We would like to thank warmly this reader for the quality of his / her  proofreading and for the relevance of his / her advice, which we have almost all strictly followed.

1- We have justified the fact that we did not quote the sociology of music (non-narrative dimension of musical works which undoubtedly hindered a sociology of musical works).

2- We have specified the contribution of L. Goldmann with a short development as was suggested.

3- We have developed a little our remarks on P. Bourdieu and on the research of G. Sapiro, especially with a reference to her recent essay Peut-on dissocier l’œuvre de l’auteur ?

4- However, we decided to keep the short development on the interpretation of inequality of opportunity in the text, because the two other reviewers did not suggest that we should remove it.

Round 2

Reviewer 1 Report

While I appreciate the responses of the authors to my comments in the first review, I remain skeptical about this text. The authors have taken up three of my suggestions which seems helpful. However, they have refrained from adjusting the structure of the text. As a result, to me it is not clear to what discourse the article contributes what points. This was my main criticism.

Also, I do not think the addition of one reference to Luhmann suffices, particularly because my comment was to his distinction between the form of communications and the Information they contain. Particularly here, he weighs in on the debate on the necessity to include the aesthetics of art works and therefore how audiences interpret them in the sociological analysis of art. As such, this presents a sociology of works of art that is more meticulously elaborated by authors such as Van Maanen. The reference to Luhmann now is only to his institutional perspective which does not differ from Bourdieu. But that was not my point. As such, the effort to connect to the German based debate is not succesful.

As a result, I remain unconvinced by this article.